# Sensitivity Assessment of Boron Isotope as Indicator of Contaminated Groundwater for Hydraulic Fracturing Flowback Fluids Produced from the Dameigou Shale of the Northern Qaidam Basin

**Zhaoxian Zheng [1,2], Yan Zhang [3] and Bingyan Li [1,2,*]**

1   Institute of Hydrogeology and Environmental Geology, Chinese Academy of Geological Sciences, Shijiazhuang 050061, China
2   Key Laboratory of Groundwater Sciences and Engineering, Ministry of Natural Resources, Shijiazhuang 050061, China
3   Geological Survey of Jiangsu Province, Nanjing 210018, China
*   Correspondence: libingyan@mail.cgs.gov.cn

**Abstract:** Hydrogeochemical processes occurring in contaminated groundwater and aquifer systems may reduce the sensitivity of boron isotopes as an indicator of hydraulic fracturing flowback fluids (HFFF) in groundwater. In this paper, based on the Chaiye-1 well (the first continental shale gas well in the northern Qaidam Basin), the hydrogeochemical processes affecting boron isotopes were analyzed in HFFF-contaminated Neogene (NG1 and NG2) and Quaternary (QG1) groundwater around the shale gas field. Then, a model for boron isotopes in HFFF-contaminated groundwater was constructed to assess the sensitivity of boron isotopes as an HFFF indicator. The results show that, limited by the range of pH values and saturation indices (SI) in HFFF-contaminated groundwater, the dissolution of alkali feldspar and precipitation of carbonate have little effect on the boron isotopes in shallow groundwater. For the NG2 aquifer system containing clay minerals, the $\delta11B$ of simulated contaminated groundwater (40.0–55.6‰) is always higher than that of the corresponding groundwater mixed conservatively (−6.4–55.6‰) due to preferential adsorption of boron isotopes onto clay minerals, indicating preferential adsorption would reduce the sensitivity of boron isotopes as an indicator of groundwater contamination from HFFF. For the scenario of HFFF contamination, when the mixing ratio of HFFF in contaminated groundwater increases by 5%, boron isotopes in Neogene (NG1 and NG2) and Quaternary (QG1) groundwater have detectable responses to HFFF contamination, suggesting $\delta11B$ is a sensitive indicator of HFFF contamination in shallow groundwater from the Dameigou Shale in the northern Qaidam Basin.

**Keywords:** shale gas; hydraulic fracturing flowback fluids; groundwater contamination; monitoring indicator; boron isotopes

## 1. Introduction

Shale gas has attracted global attention due to its high energy efficiency and cleanliness as a result of the successful application of high-volume hydraulic fracturing (HVHF) [1,2]. Hydraulic fracturing flowback fluids (HFFF), which are produced by ubiquitous HVHF, contain human-made additives and naturally occurring chemicals at toxic concentrations [3–5]. The toxic substances in HFFF may contaminate the shallow groundwater under the following contingencies: (1) leakage from a well pipe through a faulty well casing such as a corroded and poorly joined casing during flowback [6]; and (2) accidental release during storage, transport, and disposal of HFFF after flowback [7]. Consequently, potential environmental impacts on groundwater by these toxic substances have raised public concerns [8–12], and forensic identification of HFFF contamination in groundwater has become a research hotspot [13–20].

Multiple studies have focused on the isotopic fingerprint characteristics of HFFF and proposed that $\delta^2$H-H$_2$O, $\delta^{18}$O-H$_2$O, $\delta^{34}$S-SO$_4$, $\delta^{18}$O-SO$_4$, $^{87}$Sr/$^{86}$Sr, and $^{228}$Ra/$^{226}$Ra can be tracers for HFFF [18,21,22]. Among them, based on the mechanisms of hydrogeochemical formation of HFFF, the boron isotope ratio ($\delta^{11}$B) was considered to provide a novel diagnostic signature for characterizing HFFF and distinguishing HFFF from natural water [7,18,22]. Warner et al. [23] stated that $\delta^{11}$B values for HFFF from the Marcellus and Fayetteville Formations (marine shales) fall within a narrow range (25 to 31‰) with high B/Cl ratios (>0.1 $\times$ 10$^{-1}$), and were distinct from major river systems ($\delta^{11}$B = 3 to 14‰, B/Cl = 0.2 $\times$ 10$^{-4}$–0.1 $\times$ 10$^{-1}$) and shallow groundwater ($\delta^{11}$B = 34 to 54‰, B/Cl < 0.3 $\times$ 10$^{-4}$) in shale gas fields. Ni et al. [24] reported that the HFFF from the Permian Longmaxi Formation in the Sichuan Basin ($\delta^{11}$B = 22.5 to 31.6‰) had similar $\delta^{11}$B values to those of HFFF from the Marcellus Formation and were different from shallow groundwater ($\delta^{11}$B = −7.8 to 6.4‰) [25]. Additionally, Cui et al. [16] proposed that extensive dissolution of feldspars occurring during hydraulic fracturing can impart a unique $\delta^{11}$B fingerprint from continental shale (−30.1 to 10.2‰) to HFFF. Thus, $\delta^{11}$B can be reliably used in North America and Asia to identify the release of HFFF to rapidly circulating shallow groundwater that is mainly recharged by rainwater ($\delta^{11}$B: −1.5 to 34.7‰). These studies have demonstrated that $\delta^{11}$B would be a useful indicator of HFFF contamination in shallow groundwater; however, few studies have focused on hydrogeochemical processes that may cause boron isotope fractionation and the mixing of exogenous boron when HFFF enter the aquifer. The disruption of the original water chemical balance in aquifer systems resulting from HFFF contamination can lead to a series of water–rock interactions [26,27]. The adsorption of boron onto clay [28,29] and precipitation of boron during neoformation of secondary phases [30] can affect the equilibrium exchange of B(OH)$_3$ and B(OH)$_4$$^-$ in aquifers and change the relative content of boron in different structural phases, resulting in fractionation of boron isotopes [31,32]. Dissolution can also release boron from the mineral lattice and change the $\delta^{11}$B in contaminated groundwater [27,33]. Thus, these processes may reduce the sensitivity of $\delta^{11}$B as an HFFF indicator in groundwater and may even negate $\delta^{11}$B as an HFFF indicator entirely.

The aim of this study is to assess the sensitivity of $\delta^{11}$B as an HFFF indicator through a better understanding of the effects of hydrogeochemical processes on boron isotope evolution in HFFF-contaminated shallow groundwater. Our previous study reported that $\delta^{11}$B of HFFF produced from the Dameigou Shale in the northern Qaidam Basin (−10.2 to −6.4‰) was lower than and distinct from that of the shallow groundwater (24.4 to 55.7‰) [16,34]. In this study, we focus on variations in boron concentrations and isotopes affected by hydrogeochemical processes in HFFF-contaminated shallow aquifers using a quantitative hydrogeochemical model. The results of this study provide a methodology for using boron isotopes to identify groundwater contaminated by HFFF produced from the Dameigou Shale in the northern Qaidam Basin.

## 2. Materials and Methods

### 2.1. Study Area

The study area is located within the Yuka Sag in front of the Qilian Mountains in the North Qaidam Margin block fault zone (Figure 1). The study area has a typical continental plateau climate characterized by drought. The average annual precipitation is only 89.4 mm, while the average annual evaporation is as high as 2167.1 mm, more than 24 times the precipitation. The Yuka Depression, which is surrounded by mountains on three sides and faces the river on one side, is relatively closed. The main landform types in the area are structurally denuded low mountains, river terraces, riverbeds, and other fluvial landforms. The shallow aquifer in the area is mainly composed of Quaternary Upper Pleistocene gravel and Neogene siltstone. Groundwater flow directions in the study area are controlled by topography and are generally consistent with the flow directions of rivers. The Chaiye-1 well (CY1), the first continental shale gas well in northwest China, is located in the river terrace of the Naoer River. Furthermore, the Dameigou Shale is the target stratum of the

CY1 well [33]. The CY1 well, with a depth of 2250 m, was fractured in 2014. In addition, the Dameigou Shale (depth of 2000 m to 2045 m) is the target stratum of the fracturing [35]. The main components of fracturing fluid are slick water, gel liquid, and hydrochloric acid. The volume of fracturing fluid utilized for the CY1 vertical well was 890 m$^3$ for the purpose of shale gas reservoir parameter evaluation, and the cumulative volume of flowback fluid was 765 m$^3$ after this fracturing.

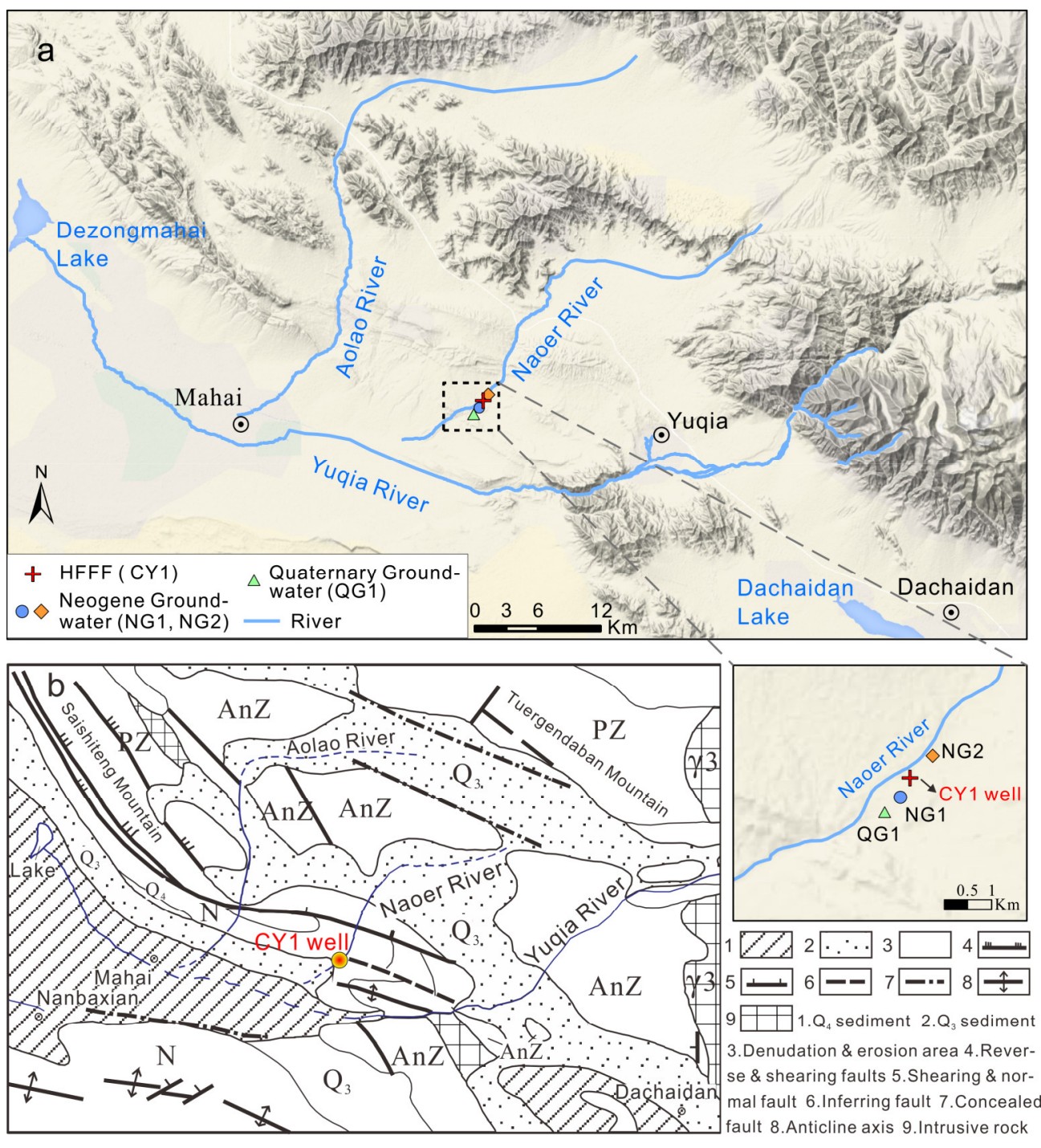

**Figure 1.** (**a**) The sampling locations of the CY1 well and shallow groundwater (NG1, NG2, and QG1) are labeled on the topographic map of the study area (**b**) Geologic map of the study area.

### 2.2. Sample Collection

Two types of water were collected in the study area: (1) five samples of HFFF produced from CY1 (CY1-1d to CY1-7d) were collected as a time series from the first to the last day of the artesian flowback process, and (2) two samples of shallow groundwater (NG1 and NG2) and one sample of spring water (QG1) from the Neogene and Quaternary aquifers, respectively, were collected before hydraulic fracturing in order to characterize the background hydrochemical compositions of groundwater. The HFFF sampling days during the flowback process are shown in Table 1. NG1 and NG2 represent the groundwater from the two major shallow Neogene aquifers in the study area. QG1 represents the groundwater from the major shallow Quaternary aquifer around the CY1 shale gas field. The depths of the NG1, NG2, and QG1 aquifers are 145.0 to 150.5 m, 96.2 to 105.0 m, and 0 to 14.5 m, respectively.

**Table 1.** Hydrochemical and isotopic results of the CY1 HFFF and shallow groundwater before fracturing.

| Sample ID | Description | Na | K | Ca | Mg | Cl | HCO$_3$ | SO$_4$ | TDS | B | $\delta^{11}$B (‰) | pH | T (°C) |
|---|---|---|---|---|---|---|---|---|---|---|---|---|---|
| | | | | | | mg/L | | | | | | | |
| CY1-1d | HFFF, flowback day 1 | 1683 | 68 | 375 | 239 | 3486 | 415 | 412 | 6413 | 1.26 | −9.6 | 6.33 | 29.1 |
| CY1-2d | HFFF, flowback day 2 | 2815 | 77 | 410 | 170 | 4783 | 1368 | 90 | 8973 | 1.04 | −10.2 | 7.00 | 38.5 |
| CY1-3d | HFFF, flowback day 3 | 2897 | 80 | 359 | 162 | 4662 | 1602 | 89 | 8990 | 1.01 | −7.1 | 7.02 | 40.1 |
| CY1-5d | HFFF, flowback day 5 | 3263 | 83 | 316 | 129 | 5042 | 1688 | 41 | 9658 | 0.92 | −8.3 | 7.73 | 43.6 |
| CY1-7d | HFFF, flowback day 7 | 3467 | 89 | 294 | 113 | 5228 | 1764 | 46 | 10,056 | 0.86 | −6.4 | 7.89 | 46.1 |
| NG1 | Groundwater, Neogene aquifer 1 | 3209 | 13 | 733 | 361 | 6014 | 33 | 1328 | 11,675 | 2.31 | 26.1 | 7.23 | 14.3 |
| NG2 | Groundwater, Neogene aquifer 2 | 4044 | 34 | 723 | 568 | 6555 | 100 | 3113 | 15,088 | 2.65 | 55.7 | 7.29 | 14.8 |
| QG1 | Groundwater, Quaternary aquifer | 1796 | 66 | 224 | 329 | 2017 | 370 | 2052 | 6791 | 4.41 | 24.4 | 8.08 | 14.6 |

In the field, water samples were filtered through a 0.45 μm nylon filter and then collected into sterilized HDPE bottles. Samples for trace element analyses were acidified to pH 2 with ultrapure nitric acid (HNO$_3$). The collected water samples were stored on ice while in the field and refrigerated in the laboratory at 4 °C until the analyses were completed.

### 2.3. Sample Analysis

Major anions were measured using ion chromatography (IC) via a Thermo Scientific Dionex ICS-4000 (precision ± 1%), except for HCO$_3^-$. The HCO$_3^-$ concentration was determined by phenolphthalein titration. Major cations and minor elements were measured using inductively coupled plasma optical emission spectroscopy (ICP-OES) via a PerkinElmer Model Optima 8300 (precision ± 1%). Boron isotopes were analyzed using inductively coupled plasma mass spectrometry (ICP-MS) via a PerkinElmer ELAN DCR-e. The average $^{11}$B/$^{10}$B ratio of NIST SRM 951 during this study was 4.0436 ± 0.0016 (*n* = 25). The major ions and minor elements were measured at the National Research Center for Geoanalysis, Chinese Academy of Geological Sciences. Boron isotopes were analyzed at the Center of Analysis, Beijing Research Institute of Uranium Geology.

## 3. Results and Discussion

### 3.1. Characteristics of Hydrochemical Compositions and Boron Isotope in HFFF and Shallow Groundwater

The hydrochemical and isotopic results of the HFFF and shallow groundwater analyses are shown in Table 1. HFFF are characterized by high total dissolved solids (TDS) (6.4 to 10.1 g/L), are weakly alkaline (6.3 to 7.89), and are dominant in Cl$^-$ and Na$^+$. Similarly, the shallow groundwater samples are also characterized by high TDS (6.8 to 15.1 g/L), are weakly alkaline (7.23 to 8.08), and are dominant in Cl$^-$ (or Cl$^-$ and SO$_4^{2-}$) and Na$^+$. Because the TDS of HFFF produced from Dameigou continental shale are significantly lower

than those of HFFF produced from marine shales such as the Marcellus Formation in the United States (14.8 to 211.4 g/L) [36], the Bowland Formation in the United Kingdom (62.6 to 99.5 g/L) [37], the Longmaxi Formation in China (13.1 to 53.5 g/L) [38], and Polish shale formations (103.2 g/L) [39], and because the dominating ions of shallow groundwater are similar to those of HFFF in this study, the conventional tracers of HFFF such as TDS, Cl, and Na/Cl are not applicable to identifying groundwater contamination by HFFF in the study area. The $SO_4^{2-}$ concentrations of the Dameigou Shale HFFF, except for the first flowback day (41 to 90 mg/L), are slightly higher than those reported for the Marcellus Formation (1.3 to 11.7 mg/L) [40] and Longmaxi Formation (23 to 89 mg/L) [41], but are lower than those reported for the Bowland Formation (101.8 to 120.1 mg/L) [37]. The $SO_4^{2-}$ concentrations of shallow groundwater in the study area (1328 to 3113 mg/L) were much higher than those of the Dameigou Shale HFFF (41 to 412 mg/L), while the $HCO_3^-$ concentrations of shallow groundwater (33 to 370 mg/L) were much lower than those of the Dameigou Shale HFFF (415 to 1764 mg/L). These results reflect that the saturation/unsaturation states of the main mineral components in HFFF and shallow groundwater are different from each other. Using pH values, temperature, and concentrations of major ions, the saturation index (SI) values of HFFF and shallow groundwater were calculated (Table 2). The SI values of gypsum, anhydrite, halite, and sylvite for the HFFF (−2.75 to −1.01, −2.89 to −1.26, −3.97 to −3.52, and −4.95 to −4.77, respectively) and shallow groundwater (−0.06 to −0.50, −0.95 to −0.50, −4.16 to −3.35, and −5.39 to −4.96, respectively) were less than 0, indicating that these minerals were in the unsaturated state in both the HFFF and shallow groundwater. Considering the pH and temperature changes in the contaminated groundwater, the listed minerals are always in the unsaturated state in contaminated groundwater, even at different degrees of contamination.

**Table 2.** The saturation index (SI) values of carbonates, sulfates, and chloride-type salts in the CY1 HFFF and shallow groundwater.

| Sample ID | Calcite (CaCO$_3$) | Aragonite (CaCO$_3$) | Gypsum (CaSO$_4$·2H$_2$O) | Anhydrite (CaSO$_4$) | Halite (NaCl) | Sylvite (KCl) |
|---|---|---|---|---|---|---|
| CY1−1d | −0.17 | −0.32 | −1.01 | −1.26 | −3.97 | −4.95 |
| CY1−2d | 1.15 | 1.01 | −1.72 | −1.87 | −3.64 | −4.83 |
| CY1−3d | 0.98 | 0.85 | −2.75 | −2.89 | −3.63 | −4.83 |
| CY1−5d | 1.74 | 1.61 | −2.17 | −2.28 | −3.56 | −4.80 |
| CY1−7d | 1.90 | 1.77 | −2.06 | −2.14 | −3.52 | −4.77 |
| NG1 | −0.57 | −0.72 | −0.31 | −0.76 | −3.47 | −5.39 |
| NG2 | −0.12 | −0.27 | −0.06 | −0.50 | −3.35 | −4.96 |
| QG1 | 0.84 | 0.69 | −0.50 | −0.95 | −4.16 | −5.11 |

The boron concentrations of the HFFF (0.86 to 1.26 mg/L) decreased with the flowback day and were slightly lower than those of shallow groundwater (2.31 to 4.41 mg/L). Meanwhile, the $\delta^{11}B$ of the HFFF (−10.2 to −6.4‰) increased with flowback day and are significantly lower than those of shallow groundwater (24.4 to 55.7‰) and those of HFFF produced from marine shales such as the Marcellus Formation (25 to 31‰) [23] and Longmaxi Formation (22.5 to 31.6‰) [24]. These indicate that the $\delta^{11}B$ of groundwater is expected to have a significant response to continental HFFF contamination in the study area. The boron isotope characteristics of the HFFF were mainly evolved by the mixing of a shale formation water in which dissolution of non-marine borates (−30.1 to 10.2‰ with mean value of 4‰) has occurred during the formation of the Dameigou Shale, and the water–rock interactions were dominated by the dissolution of boron-bearing alkali feldspars during hydraulic fracturing and subsequent flowback periods [42]. Conversely, the boron isotope characteristics of the shallow groundwater were mainly imparted by the local rainwater ($\delta^{11}B$ = 16.7‰), which is the major source of groundwater recharge, and groundwater in specific aquifers (NG2) was further altered by clay adsorption [43,44]. Based on the different formation mechanisms and distinctive values of $\delta^{11}B$ in the HFFF

and shallow groundwater, boron isotopes can be used as tracers for HFFF contamination in shallow groundwater.

### 3.2. Hydrogeochemical Processes Affecting Boron Isotopes in HFFF-Contaminated Aquifers

Hydrogeochemical processes, which would occur in HFFF-contaminated aquifer systems induced by disruption of the original water chemical balance, may be the main factors significantly affecting boron isotopes in contaminated groundwater, in addition to conservative mixing of groundwater and HFFF. These hydrogeochemical processes can affect the equilibrium and cause dynamic fractionation of boron isotopes in aquifer systems, and also influence boron isotopes largely through the dissolution of boron-containing aquifer media. Because boron has one main oxidation state (+3, i.e., $B^{3+}$) and does not exist in natural water in the gaseous phase, redox reactions, microbial metabolism, and physical actions such as evaporation and volatilization do not result in boron isotope fractionation [45]. Therefore, the following hydrogeochemical processes, other than the conservative mixing of HFFF and groundwater, that may further modify $\delta^{11}B$ of contaminated groundwater were considered in this study:

### 3.2.1. Dissolution of Alkali Feldspars

During the mineralization process, $B^{3+}$ can be trapped in the alkali feldspars (albite and K-feldspar) by substitution with $Si^{4+}$ or $Al^{3+}$, thus forming boron silicate compounds [46]. Because albite and K-feldspar are present in both the Neogene and Quaternary aquifers in the study area, it is necessary to consider the boron from incongruent dissolution of alkali feldspars in contaminated groundwater [33]. The conservative mixing model for groundwater and HFFF hydrogeochemical compositions was established using the Mixing Module of PHREEQC (Version 3.4.0, USGS, Reston, VA, USA) in order to assess the stability of alkali feldspars in the contaminated aquifer system. The CY1-7d HFFF sample, which has smaller differences in $\delta^{11}B$ from the background groundwater, was selected as the endmember of the contamination source.

Based on thermodynamic reactions, the contaminated groundwater is in the stability field of K-feldspar only when the HFFF flux reaches more than approximately 30% of groundwater (NG1, NG2, and QG1), as shown in the mineral stability diagram for K-feldspar (Figure 2a–c). Also, the contaminated groundwater is always in the instability field of albite, regardless of the flux ratio of HFFF in groundwater (Figure 2d–f). These data suggest that alkali feldspar dissolution would occur in both the Neogene and Quaternary aquifers in the study area. On the basis of chemical kinetics, Chou and Wollast [47] reported that the steady-state dissolution rates of albite and K-feldspar in hydrochloric acid at 25 °C and pH 6 to 8 are about $0.1 \times 10^{-12}$ mol/m$^2$/s and $0.5 \times 10^{-14}$ mol/m$^2$/s, respectively. Moreover, Lasaga [48] stated that the dissolution half-life for a 1 mm crystal of K-feldspar is $5.2 \times 10^5$ y at 25 °C and pH 5, and the half-life would be longer in the pH range of 6 to 8. These studies demonstrate that alkali feldspars in both the Neogene and Quaternary aquifers contaminated by HFFF dissolves at very low rates because the pH variations in the contaminated groundwater samples NG1, NG2, and QG1 are in the range of 7.23 to 7.89, 7.28 to 7.89, and 7.89 to 8.08, respectively. Therefore, under the premise of dynamic groundwater monitoring with a frequency of three months, which is common in shale gas fields, too little boron is released from the dissolution of alkali feldspars in the aquifers to affect the boron isotopes in HFFF-contaminated groundwater.

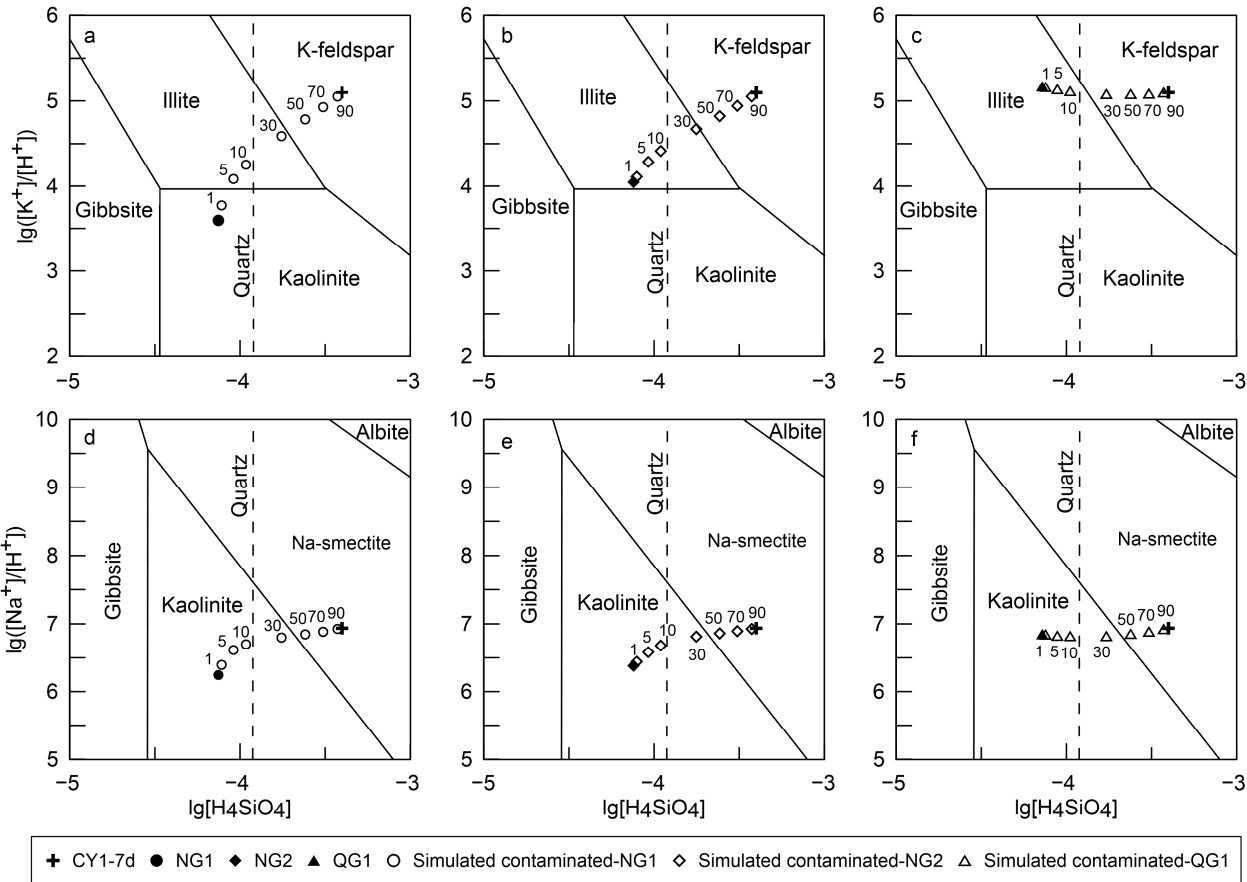

**Figure 2.** Mineral stability diagrams in weathering systems of (**a–c**) K-feldspar and (**d–f**) albite for the simulated NG1, NG2, and QG1 groundwater contaminated by 7th day HFFF, respectively. The equilibrium phase boundary was plotted at 15 °C and 0.1 MP. The labeled tick marks on the curve indicate the mixing ratio of HFFF (in volume percentage) in contaminated groundwater.

### 3.2.2. Carbonate Precipitation

Borate ions ($^{10}B(OH)_4{}^-$) can be incorporated into calcium carbonate ($CaCO_3$) lattices through carbonate precipitation, resulting in the kinetic fractionation of boron isotopes and an increase in $\delta^{11}B$ in aqueous solutions [32,49]. The results of conservative mixing show that the SI values of calcite and aragonite are greater than 0 in the contaminated groundwater samples NG1, NG2, and QG1 when the HFFF flux reaches more than 1% of the groundwater, indicating that carbonates in both the contaminated Neogene and Quaternary aquifers are in the saturation state. However, previous studies reported that carbonates precipitate in aqueous solution only when the saturation index is greater than 2 or 2.5 [50]. Consequently, boron isotope fractionation caused by carbonate precipitation in contaminated aquifer systems does not need to be considered in this study.

### 3.2.3. Boron Adsorption of Clay Minerals

In aqueous solutions, boron is mainly present in the form of boric acid ($B(OH)_3$). When the pH of the aqueous solution is higher than 6.7, $B(OH)_4{}^-$ is generated through the hydrolysis of $B(OH)_3$, as shown in Equation (1), and the reaction process is mainly controlled by the pH [51]. The hydrolysis of $B(OH)_3$ is accompanied by equilibrium fractionation of boron isotopes [52]. Because the fractionation factor of $B(OH)_3$ is always higher than that of $B(OH)_4{}^-$ under different temperature conditions, $B(OH)_3$ is always preferentially enriched in $^{11}B$, and $B(OH)_4{}^-$ is enriched in $^{10}B$ (Equation (2)) [31,53]. Thus, the $\delta^{11}B$ of contaminated groundwater can be affected through selectively adsorbing $^{10}B(OH)_4{}^-$ in the liquid phase onto clay minerals in the aquifers.

The aquifers represented by NG1 and QG1 consists of quartz, feldspar, and biotite, while clay minerals are absent, which plays a key role in boron isotope equilibrium fractionation. Hence, little boron isotope fractionation would be caused by adsorption between the solid and aqueous phases after HFFF leakage into the aquifers at NG1 and QG1. Therefore, the $\delta^{11}B$ of contaminated groundwater in the NG1 and QG1 aquifers can be generalized as conservative mixing. However, the aquifer represented by NG2 includes approximately 10% clay minerals. These minerals could cause equilibrium fractionation of boron isotopes between the aqueous and solid phases with preferential adsorption of $^{10}B(OH)_4^-$, which was influenced by the variation of pH values (from 7.29 to 7.89, Table 3) resulting from increasing HFFF in contaminated groundwater. Thus, the effect of adsorption must be considered in the NG2 aquifer system.

$$B(OH)_3 + H_2O \rightarrow B(OH)_4^- + H^+ \tag{1}$$

$$^{10}B(OH)_3 + {^{11}B(OH)_4^-} \rightarrow {^{11}B(OH)_3} + {^{10}B(OH)_4^-} \tag{2}$$

**Table 3.** Summary of parameters and process data used and calculated for $\delta11B$ in contaminated NG2 groundwater, considering equilibrium fractionation caused by preferential adsorption.

| Mixing Ratio of CY1−7d in Contaminated Groundwater | | 0.0% | 1.0% | 5.0% | 10.0% | 30.0% |
|---|---|---|---|---|---|---|
| NG1 | $[B]_{CMW}$ (mM) | 0.214 | 0.213 | 0.207 | 0.201 | 0.174 |
| | $\delta^{11}B_{CMW}$ (‰) | 26.1 | 26.0 | 25.5 | 24.8 | 21.7 |
| | $\Delta[B]_{contamination}$ (mM) | / | −0.001 | −0.007 | −0.013 | −0.04 |
| | $\Delta\delta^{11}B_{contamination}$ (‰) | / | −0.1 | −0.6 | −1.3 | −4.5 |
| | $SI_{Calcite}$ | −0.57 | −0.22 | 0.38 | 0.69 | 1.17 |
| | $SI_{Aragonite}$ | −0.72 | −0.37 | 0.23 | 0.54 | 1.01 |
| QG1 | $[B]_{CMW}$ (mM) | 0.408 | 0.405 | 0.392 | 0.375 | 0.310 |
| | $\delta^{11}B_{CMW}$ (‰) | 24.4 | 24.3 | 24.0 | 23.7 | 22.0 |
| | $\Delta[B]_{contamination}$ | / | −0.003 | −0.016 | −0.033 | −0.098 |
| | $\Delta\delta^{11}B_{contamination}$ | / | −0.1 | −0.3 | −0.7 | −2.4 |
| | $SI_{Calcite}$ | 0.84 | 0.85 | 0.90 | 0.95 | 1.12 |
| | $SI_{Aragonite}$ | 0.69 | 0.70 | 0.74 | 0.79 | 0.96 |
| Mixing ratio of CY1−7d in contaminated groundwater | | 50.0% | 70.0% | 90.0% | 95.0% | 100.0% |
| NG1 | $[B]_{CMW}$ (mM) | 0.147 | 0.120 | 0.094 | 0.086 | 0.080 |
| | $\delta^{11}B_{CMW}$ (‰) | 17.3 | 11.0 | 1.0 | −2.4 | −6.4 |
| | $\Delta[B]_{contamination}$ (mM) | −0.067 | −0.094 | −0.12 | −0.128 | −0.134 |
| | $\Delta\delta^{11}B_{contamination}$ (‰) | −8.8 | −15.1 | −25.1 | −28.6 | −32.6 |
| | $SI_{Calcite}$ | 1.36 | 1.46 | 1.89 | 1.89 | 1.90 |
| | $SI_{Aragonite}$ | 1.20 | 1.30 | 1.76 | 1.76 | 1.77 |
| QG1 | $[B]_{CMW}$ (mM) | 0.244 | 0.178 | 0.113 | 0.096 | 0.080 |
| | $\delta^{11}B_{CMW}$ (‰) | 19.3 | 14.7 | 4.7 | 0.1 | −6.4 |
| | $\Delta[B]_{contamination}$ | −0.164 | −0.230 | −0.295 | −0.312 | −0.328 |
| | $\Delta\delta^{11}B_{contamination}$ | −5.0 | −9.6 | −19.6 | −24.2 | −30.8 |
| | $SI_{Calcite}$ | 1.25 | 1.37 | 1.47 | 1.49 | 1.90 |
| | $SI_{Aragonite}$ | 1.10 | 1.22 | 1.32 | 1.34 | 1.77 |

Note: $\Delta\delta^{11}B_{contamination} = \delta^{11}B_{NG1\,or\,QG1} - \delta^{11}B_{CMW}$, $\Delta[B]_{contamination} = [B]_{NG1\,or\,QG1} - [B]_{CMW}$; "/" indicates the value does not need to be acquired from NG2 groundwater without mixing with HFFF.

### 3.3. Modeling of $\delta^{11}B$ in Contaminated Groundwater Considering Boron Isotope Equilibrium Fractionation Caused by Adsorption

HFFF contamination can be divided into three steps to model the $\delta^{11}B$ in contaminated groundwater considering equilibrium fractionation.

(1) Equilibrium fractionation of boron isotopes between aquifer media and groundwater occurs at a background pH of 7.29 and temperature of 14.8 °C before HFFF contamination.

The first step is expressed as:

$$Kd_B = \frac{[B]_{BS}}{[B]_{BW}}$$

(3)

$$\alpha_B = \frac{\delta^{11}B_{BS} + 1000}{\delta^{11}B_{BW} + 1000}$$

(4)

where $Kd_B$ is the background distribution coefficient of the aquifer system before contamination; $[B]_{BS}$ and $[B]_{BW}$ are the background boron concentrations of the aquifer media and groundwater, respectively; $\alpha_B$ is the background fractionation factor between the adsorbed and dissolved species of boron; and $\delta^{11}B_{BS}$ and $\delta^{11}B_{BW}$ are the background $\delta^{11}$B values of the aquifer media and groundwater, respectively.

(2) Conservative mixing of background groundwater and HFFF causes a change in $\delta^{11}$B in the aqueous phase.

The second step is expressed as:

$$[B]_{CMW} = [B]_{BW}(1 - x_{HW}) + [B]_{HFFF}x_{HW}$$

(5)

$$\delta^{11}B_{CMW} = \frac{[B]_{BW}\delta^{11}B_{BW}(1 - x_{HW}) + [B]_{HFFF}\delta^{11}B_{HFFF}x_{HW}}{[B]_{CMW}}$$

(6)

where $[B]_{CMW}$ and $\delta^{11}B_{CMW}$ are the boron concentration and $\delta^{11}$B of conservative mixing between background groundwater and HFFF before adsorption occurred in the contaminated aquifer system, respectively; $[B]_{HFFF}$ and $\delta^{11}B_{HFFF}$ are the boron concentration and $\delta^{11}$B of HFFF, respectively; and $x_{Hw}$ is the volume percentage of HFFF in the HFFF-contaminated groundwater. $[B]_{BW}$ and $\delta^{11}B_{BW}$ are the same variables as described in Equations (3) and (4).

(3) The changed pH causes hydrolysis of B(OH)$_3$ (Equation (1)), which is associated with the boron isotope exchange reaction (Equation (2)) in the aqueous phase. Then, a new equilibrium fractionation of boron isotopes between aquifer media and conservative mixing water occurs after preferential adsorption/desorption of $^{10}$B(OH)$_4$ in the aqueous phase on clay minerals under the changed pH of contaminated groundwater.

The final step is expressed as:

$$Kd_C = \frac{[B]_{CS}}{[B]_{CW}} = \frac{[B]_{BS} + [B]_{adsorbS}}{[B]_{CMW} - [B]_{adsorbedW}}$$

(7)

$$[B]_{adsorbS} = u[B]_{adsorbedW}$$

(8)

$$\alpha_C = \frac{\delta^{11}B_{CS} + 1000}{\delta^{11}B_{CW} + 1000} = \frac{\frac{\delta^{11}B_{BS} \times [B]_{BS} + \delta^{11}B_{adsorbed} \times [B]_{adsorbS}}{[B]_{BS} + [B]_{adsorbS}} + 1000}{\frac{\delta^{11}B_{CMW} \times [B]_{CMW} - \delta^{11}B_{adsorbed} \times [B]_{adsorbedW}}{[B]_{CMW} - [B]_{adsorbedW}} + 1000}$$

(9)

where $Kd_c$ is the distribution coefficient of the contaminated groundwater with changed pH and temperature; $[B]_{CS}$ and $[B]_{CW}$ are the boron concentrations of the contaminated aquifer media and groundwater, respectively; and $[B]_{adsorbS}$ and $[B]_{adsorbedW}$ are the elevated boron concentrations of the aquifer media absorbed from conservative mixing water and the decreased boron concentration of mixing water caused by adsorption, respectively. In addition, $u$ is the ratio of the mass of water to the mass of oven-dried material; $\alpha_C$ is the fractionation factor between the adsorbed and dissolved species of boron of the contaminated aquifer system; $\delta^{11}B_{CS}$ and $\delta^{11}B_{CW}$ are the $\delta^{11}$B values of the contaminated aquifer media and groundwater after conservative mixing and adsorption/desorption, respectively; and $\delta^{11}B_{absorbed}$ is the $\delta^{11}$B of absorbed boron from conservative mixing water to the aquifer media. $[B]_{BS}$, $[B]_{CMW}$, $\delta^{11}B_{BS}$, and $\delta^{11}B_{CMW}$ are the same variables as described in Equations (3)–(6).

### 3.4. Sensitivity Assessment of Boron Isotopes as an Indicator of HFFF-Contaminated Groundwater

To assess the sensitivity of boron isotopes as an indicator of HFFF-contaminated groundwater, the seventh-day HFFF and background groundwater (NG1, NG2, and QG1) were selected as the endmembers of the contamination source and contamination receptors, respectively. The NG1 and QG1 samples contaminated by HFFF can be generalized as examples of conservative mixing (Table 3). However, the NG2 contamination was simulated using the model established in Section 3.3. Experimental data for $K_d$ and $\alpha$ in the NG2 aquifer system containing 10% clay minerals are lacking. Instead, it was assumed that the NG2 aquifer medium is entirely composed of clay minerals, which have experimental data for $K_d$ and $\alpha$ under a variety of pH and temperature conditions. If there is little boron isotope fractionation that plausibly occurs in the environment in the range of HFFF inputs for the above assumption, $\delta^{11}B$ could be a cogent tool for identifying HFFF from NG2 groundwater under real hydrogeological conditions.

The pH values for contaminated groundwater at different HFFF mixing ratios were obtained by the Mixing Module of PHREEQC (Version 3, USGS, Reston, VA, USA). The $Kd$ and $\alpha$ values between the adsorbed and dissolved species of boron for a variety of pH values and a temperature of 15 °C can be checked from previous clay mineral experimental studies (Table 4). Then, using the collected data of $Kd_B$ and $\alpha_B$ [29] and the measured data of $[B]_{BW}$ and $\delta^{11}B_{BW}$ (Table 1) for the background equilibrium fractionation state before contamination, $[B]_{BS}$ (0.458 mM/kg) and $\delta^{11}B_{BS}$ (25.5‰) are obtained in the first step (Equations (1) and (2)) of modeling boron isotopes. The $[B]_{CMW}$ and $\delta^{11}B_{CMW}$ values can be obtained using the measured data of $[B]_{BW}$, $[B]_{HFFF}$, $\delta^{11}B_{BW}$, and $\delta^{11}B_{HFFF}$ (Table 1) at different $x_{HW}$ values in the second step (Table 4). Finally, $\delta^{11}B_{CW}$ can be calculated at the new equilibrium fractionation state after mixing and adsorption/desorption (Table 4).

The simulation results of groundwater contamination by HFFF show that $\delta^{11}B$ of the NG2 contaminated groundwater ($\delta^{11}B_{CW}$: 40.0 to 55.6‰, Table 4) in a variety of HFFF mixing ratios (1 to 100%), considering the mixing of background groundwater and HFFF and the selective adsorption by clay minerals after mixing, is always higher than those of the corresponding contaminated groundwaters mixed conservatively ($\delta^{11}B_{CMW}$:−6.4 to 55.6‰, Table 4), suggesting that preferential adsorption of clay minerals would reduce the sensitivity of boron isotopes as an indicator of HFFF contamination in groundwater. Although the $\Delta\delta^{11}B_{preferential\ adsorption}$ ($\Delta\delta^{11}B_{preferential\ adsorption} = \delta^{11}B_{CW} - \delta^{11}B_{CMW}$) gradually increases with an increasing degree of contamination (or HFFF mixing ratio), a slight increase of $\Delta\delta^{11}B_{preferential\ adsorption}$ (<3.6‰) would be caused in the range of HFFF inputs (<30% HFFF) that might plausibly occur in the environment. This indicates that selective adsorption has little impact on the usability of $\delta^{11}B$ as an indicator of HFFF in NG2 groundwater in the study area.

Response curves of the contamination degree were also generated for the NG1, NG2, and QG1 groundwater interacting with the seventh-day HFFF (Figure 3). For the scenario of HFFF contamination, a flux equivalent to 5% of NG1, NG2, and QG1 groundwater can result in detectable shifts in the $\delta^{11}B$ ($\Delta\delta^{11}B_{contamination}$ = −0.6‰, −0.6‰, and −0.3‰, respectively) and boron concentration ($\Delta[B]_{contamination}$ = −0.07 mg/L, −0.30 mg/L, and −0.18 mg/L, respectively) for the shallow groundwater (Tables 3 and 4). When the HFFF flux reaches 30% of the NG1, NG2, and QG1 groundwater, the $\delta^{11}B$ ($\Delta\delta^{11}B_{contamination}$ = −4.5‰, −3.9‰, and −2.4‰, respectively) and boron concentration ($\Delta[B]_{contamination}$ = 0.44 mg/L, 0.66 mg/L, and 1.07 mg/L, respectively) for all shallow groundwater are significantly affected. These results illustrate that boron isotopes can be an effective indicator for identifying groundwater contaminated by HFFF produced from the Dameigou Shale in the study area.

**Table 4.** Summary of parameters and process/result data used and calculated for $\delta^{11}B$ in contaminated NG2 groundwater considering equilibrium fractionation caused by preferential adsorption at different HFFF mixing ratios.

| Mixing Ratio of CY1-7d in Contaminated NG2 Groundwater | 0.0% | 1.0% | 5.0% | 10.0% | 30.0% |
|---|---|---|---|---|---|
| pH value | 7.29 | 7.35 | 7.50 | 7.59 | 7.73 |
| $Kd_{mix}$ at 15 °C [31] | 1.868 | 1.932 | 2.123 | 2.256 | 2.503 |
| $\alpha_{mix}$ at 15 °C [31] | 0.9 714 | 0.9 719 | 0.9 729 | 0.9 734 | 0.9 742 |
| $[B]_{CMW}$ (mM) | 0.245 | 0.244 | 0.237 | 0.229 | 0.196 |
| $\delta^{11}B_{CMW}$ (‰) | 55.7 | 55.5 | 54.6 | 53.5 | 48.1 |
| $[B]_{absorbedW}$ (mM) | / | 0.006 3 | 0.020 3 | 0.024 6 | 0.012 1 |
| $\delta^{11}B_{absorbed}$ (‰) | / | 52.8 | 49.3 | 45.0 | −8.1 |
| $[B]_{CW}$ (mM) | / | 0.238 | 0.217 | 0.204 | 0.184 |
| $\delta^{11}B_{CW}$ (‰) | / | 55.5 | 55.1 | 54.5 | 51.8 |
| $\Delta\delta^{11}B_{preferential\ adsorption}$ | / | 0.1 | 0.5 | 1.0 | 3.7 |
| $\Delta[B]_{contamination}$ (mM) | / | −0.007 | −0.028 | −0.041 | −0.061 |
| $\Delta\delta^{11}B_{contamination}$ | / | −0.2 | −0.6 | −1.2 | −3.9 |
| $SI_{Calcite}$ | −0.12 | −0.2 | −0.6 | −1.2 | −3.9 |
| $SI_{Aragonite}$ | −0.27 | 0.01 | 0.36 | 0.61 | 1.08 |
| Mixing ratio of CY1-7d in contaminated NG2 groundwater | 50.0% | 70.0% | 90.0% | 95.0% | 100.0% |
| pH value | 7.79 | 7.83 | 7.87 | 7.88 | 7.89 |
| $Kd_{mix}$ at 15 °C [31] | 2.617 | 2.713 | 2.800 | 2.805 | 2.822 |
| $\alpha_{mix}$ at 15 °C [31] | 0.9 745 | 0.9 747 | 0.9 749 | 0.9 749 | 0.9 749 |
| $[B]_{CMW}$ (mM) | 0.163 | 0.129 | 0.096 | 0.088 | 0.080 |
| $\delta^{11}B_{CMW}$ (‰) | 40.4 | 28.9 | 9.4 | 2.2 | −6.4 |
| $[B]_{absorbedW}$ (mM) | −0.012 1 | −0.038 1 | −0.065 1 | −0.072 8 | −0.079 9 |
| $\delta^{11}B_{absorbed}$ (‰) | 158.6 | 100.3 | 88.3 | 86.0 | 84.5 |
| $[B]_{CW}$ (mM) | 0.175 | 0.167 | 0.161 | 0.161 | 0.160 |
| $\delta^{11}B_{CW}$ (‰) | 48.6 | 45.2 | 41.2 | 40.2 | 39.1 |
| $\Delta\delta^{11}B_{preferential\ adsorption}$ | 8.2 | 16.3 | 31.8 | 38.0 | 45.5 |
| $\Delta[B]_{contamination}$ (mM) | −0.070 | −0.078 | −0.084 | −0.084 | −0.085 |
| $\Delta\delta^{11}B_{contamination}$ | −7.1 | −10.5 | −14.5 | −15.5 | −16.6 |
| $SI_{Calcite}$ | 1.29 | 1.41 | 1.49 | 1.50 | 1.90 |
| $SI_{Aragonite}$ | 1.13 | 1.26 | 1.34 | 1.35 | 1.77 |

Note: Positive values of $[B]_{adsorbedW}$ indicate the sediment adsorbed the B from mixing water, while negative values of $[B]_{adsorbedW}$ indicate the sediment desorbed the B from exchangeable sites on the mineral surfaces to mixing water; $\Delta\delta^{11}B_{preferential\ adsorption} = \delta^{11}B_{CW} - \delta^{11}B_{CMW}$, $\Delta\delta^{11}B_{contamination} = \delta^{11}B_{NG2} - \delta^{11}B_{CW}$, $\Delta[B]_{contamination} = [B]_{NG2} - [B]_{CW}$; "/" indicates the value does not need to be acquired from NG2 groundwater without mixing with HFFF.

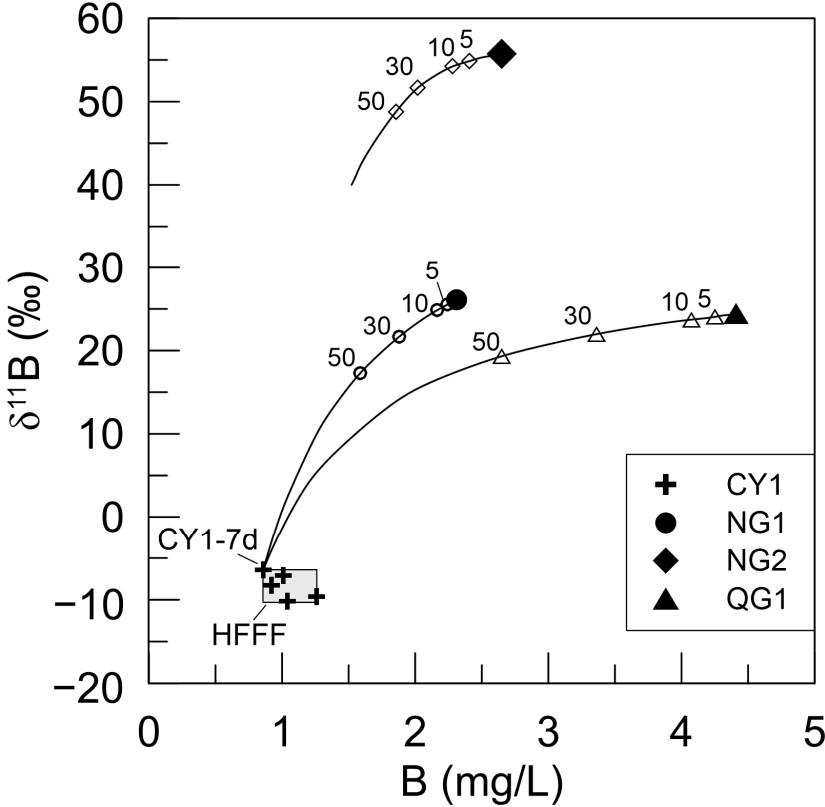

**Figure 3.** Response curves of $\delta^{11}B$ and boron concentration in NG1, NG2, and QG1 groundwater contaminated by 7th-day HFFF; the mixing ratios of the HFFF in contaminated groundwater are labeled on the response curves. The labeled tick marks on the curve indicate the mixing ratio of HFFF (in volume percentage) in contaminated groundwater.

## 4. Conclusions

In the Neogene (NG1 and NG2) and Quaternary (QG1) aquifers of the study area, the potential hydrogeochemical processes that may affect boron isotopes in HFFF-contaminated groundwater (in addition to conservative mixing of HFFF and background groundwater) are incomplete dissolution of alkali feldspar minerals and carbonate precipitation. Although albite and K-feldspar dissolution occurs in HFFF-contaminated Neogene and Quaternary aquifer systems, the very low dissolution rates of these minerals, mainly controlled by pH, result in insignificant boron being produced from dissolution, and thus this process does not affect the sensitivity of boron isotopes as an HFFF indicator in the shallow groundwater of the study area. Because carbonate is in the non-precipitating state in a range of HFFF mixing ratios, boron isotope fractionation caused by carbonate precipitation in contaminated aquifer systems does not need to be considered for this study area.

The simulation of $\delta^{11}B$ for HFFF-contaminated groundwater shows that preferential adsorption onto clay minerals in aquifer media affects boron isotopes, reducing the sensitivity of $\delta^{11}B$ as an indicator of HFFF. However, for this study area, the usability of $\delta^{11}B$ as an HFFF indicator is likely to be unaffected for the range of HFFF inputs that might plausibly occur in the environment (<30% HFFF).

For the scenario of HFFF contamination, when the mixing ratio of HFFF in contaminated groundwater increased by 5%, boron isotopes in Neogene and Quaternary groundwater had a detectable response to HFFF contamination, suggesting $\delta^{11}B$ is a sensitive indicator of HFFF contamination in the shallow groundwater from the Dameigou Shale in the northern Qaidam Basin.

**Author Contributions:** Conceptualization, Z.Z. and B.L.; methodology, Z.Z. and Y.Z.; software, Z.Z. and B.L.; validation, Y.Z. and B.L.; investigation, Z.Z.; writing—original draft preparation, Z.Z.; writing—review and editing, Z.Z. and B.L.; visualization, Z.Z.; supervision, Y.Z. All authors have read and agreed to the published version of the manuscript.

**Funding:** This research was funded by the National Natural Science Foundation of China (Grant No. 41302192); the China Geological Survey (Grant No. DD20230076); the Natural Science Foundation of Hebei Province of China (Grant No. D2018504011); and the Ministry of Land and Resources of the People's Republic of China (Grant No. 201411052).

**Data Availability Statement:** The data presented in this study are available on request from the corresponding author.

**Acknowledgments:** The authors gratefully acknowledge many important contributions from the researchers of all reports cited in our paper.

**Conflicts of Interest:** The authors declare no conflict of interest.

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
