# Peer review of "Sensitivity Assessment of Boron Isotope as Indicator of Contaminated Groundwater for Hydraulic Fracturing Flowback Fluids Produced from the Dameigou Shale of the Northern Qaidam Basin"

_sustainability, doi:10.3390/su15065481_

Round 1

Reviewer 1 Report

In this paper, based on the Chaiye-1 well (the first Continental shale gas well in the northern Qaidam Basin), the hydrogeochemical processes affecting boron isotopes were analyzed in HFFF-contaminated Neogene (NG1 and NG2) and Quaternary (QG1) groundwater around the shale gas field. This study is relatively novel and the experiments were carried out effectively, but I have the following questions:

(1) Shale gas reservoirs are usually buried at a depth of 2000-5000m strata and the fracturing fluid return fluid is returned to the surface through the wellbore. Due to the barrier effect of the casing, it is impossible for the rejection fluid to contact with shallow formation water or rock, is the experimental assumption in this paper reasonable?

(2) The shale gas well site has a special fracturing fluid collection pond, which has also been specially treated for impermeability and will not come into contact with shallow formation water.

(3) The reasonableness of the selection of index parameters and the range of values deserve further explanation and clarification. "... . δ11B of contaminated groundwater (40.0 ‰ ~ 55.6 ‰) is always higher than that of corresponding groundwater mixing conservatively (-6.4 ‰ ~ 55.6 ‰) due to preferential adsorption of boron isotopes on clay minerals..."

Reviewer 2 Report

Sensitivity assessment of boron isotope as indicator of contaminated groundwater for hydraulic fracturing flowback fluids were studied. It was atrctive.

While, the sampling depth should be introduced. Some discussion about the similar studies should be added and discussed. 

Reviewer 3 Report

Is it possible to add such information as:

1.       The well depth of Chaiye-1

2.       Composition of fracking fluids

3.       The volume of fracking fluid utlilized for Chaiye-1 hydraulic fracturing

Is it possible to add to the existing manuscript some information on other isotopes used as markers?

Eg.Quantifying the extent of flowback of hydraulic fracturing fluids using chemical and isotopic tracer approaches, F. Osselin, M. Nightingale, G. Hearn, W. Kloppmann, E. Gaucher, C.R. Clarkson, B. Mayer

Is it possible to compare flowback composision from different countries (of course according to the rock type) or just to give citation that is similar or not, and why?

1.       Guar Gum Stimulates Biogenic Sulfide Production in Microbial Communities Derived from UK Fractured Shale Production Fluids; Lisa Cliffe, Natali Hernandez-Becerra, Christopher Boothman, Bob Eden, Jonathan R. Lloyd, Sophie L. Nixona

2.       Purification of flowback fluids after hydraulic fracturing of Polish gas shales by hybrid methods; Anna Abramowska,Dorota K. Gajda, Katarzyna Kiegiel, Agnieszka MiÅ›kiewicz, PrzemysÅ‚aw Drzewicz &Grażyna Zakrzewska-KoÅ‚tuniewicz

The result of the research is valuable and I would like to recommend the manuscript for printing, however the additional data listed above will be welcome.

Round 2

Reviewer 1 Report

The rework was carefully added and improved, and I have no other questions and agree to publish it